# A New Conceptual Design of Twisting Morphing Wing

**DOI:** 10.3390/biomimetics10020110

**Published:** 2025-02-12

**Authors:** Noppawit Kumkam, Napat Suratemeekul, Suwin Sleesongsom

**Affiliations:** Department of Aeronautical Engineering, International Academy of Aviation Industry, King Mongkut’s Institute of Technology Ladkrabang, Bangkok 10520, Thailand

**Keywords:** twisting wingtip, morphing wing, aeroelasticity, UAV, Gust Alleviation Device

## Abstract

This research aims to enhance the performance of unmanned aerial vehicles (UAVs) by investigating the impact of twisting wingtip (TWT) on UAVs’ wing aeroelastic and structural behavior using MATLAB and ANSYS simulations. The study focuses on a simplified twisting wingtip design and its aeroelastic effect. This study includes both static and dynamic aeroelastic phenomena. Previous research has primarily focused on only flutter speed while neglecting divergence speed and lift-effectiveness in design results. Numerical and experimental validation underscores the model’s fidelity and its practical applicability. The TWT is designed to exhibit a predominant torsional mode using a guide mode preference technique. The design results reveal that the twist morphing wing improves structural and aeroelastic performance due to its unique twisting deformation capabilities. Furthermore, this research contributes fundamental insights into a specific twist morphing wing concept, highlighting its potential to enhance UAV performance through twisting wingtip technologies. The torsional mode can be predetermined using the guide mode preference technique. Notably, the divergence speed analysis confirms that the twisting shaft position should not exceed the aerodynamic center, which is located at 0.2103 of the chord length. This serves as the theoretical foundation for the TWT design in this study. The adjustment of the TWT’s twisting angle is confirmed to provide optimal divergence speed improvement within a range of 0% to 27.7%. Additionally, the relative aeroelastic efficiencies indicate that the highest lift effectiveness is 0.68% at a twisting angle of 30°, following an exponential relationship, which can be further extended to aircraft control laws. However, the relative efficiency of flutter speed is not significantly improved by the TWT, showing only a marginal improvement of 0% to 1.84% when twisting up and down, in accordance with previous research findings.

## 1. Introduction

A morphing wing, also known as an adaptive wing or shape-changing wing, refers to a concept in aircraft design where the shape and configuration of the wing can be continuously altered or adapted during flight. It involves the ability to change the wing’s shape, surface area, or other parameters to optimize aerodynamic performance and efficiency [1,2,3,4,5,6,7]. Twist morphing (TM) wings refer to a type of aircraft wing design that incorporates the ability to change or adjust the twist angle of the wing’s airfoil section along its span [1,6]. The alternative structure includes both unconventional ribs and spars [8,9], and rod ribs [10]. Flight tests have shown that twist morphing is an excellent strategy to command roll maneuvers. The resulting vehicles are relatively easy to fly and consequently are suitable for autopilot design and mission deployment [11]. Twisting of the wingtip is a technique employed in aircraft design where the angle of incidence or the geometric angle of attack of the wing changes along its span. This variation in wing twist, from the root to the tip, can have significant effects on the aircraft’s aerodynamic performance and overall efficiency. Twisting of wingtip is a design strategy that offers numerous benefits for aircraft performance, including reduced drag, improved lift distribution, enhanced maneuverability, and increased efficiency. By tailoring the wing twist to suit different flight conditions, aircraft designers can optimize performance across a wide range of scenarios, making this technique a valuable tool in modern aviation [12]. Another kind of TM is in the form of various active and passive control technologies that have been developed for flutter suppression and gust alleviation, with some having been applied to aircraft [13,14,15]. When comparing passive control systems with active control systems, passive devices are typically simpler and more reliable. The current investigation focuses on evaluating the effectiveness of gust alleviation using a specially designed passive twist wingtip (PTWT). Previous research has examined this specific PTWT in the context of large unmanned air vehicles, including a flying-wing aircraft [13,15], with very flexible and high aspect ratio wings. The results have consistently shown a significant reduction in gust-induced loads when employing this passive gust load alleviation device with optimized design parameters. A PTWT is a design feature in commercial aviation that involves the natural or passive variation of wingtip twist along the span of an aircraft’s wing. This twist results in a negative angle of attack (AoA) and a negative aerodynamic force on the PTWT, which contributes to gust alleviation of the wing. The Passive Gust Alleviation Device (PGAD) has been introduced to the PTWT system, which is mounted at the wingtip of a flying-wing aircraft to minimize gust-induced responses. The concept of PTWT has been illustrated in [14], where a separate rigid wing section, known as the TWT, is mounted at the wingtip through an elastic hinge. This elastic hinge comprises a torque spring and a rotation shaft, as depicted in [13,14,15]. By positioning the axis of the rotation shaft in front of the aerodynamic center, the TWT twists downward to mitigate the aerodynamic forces in response to gust loads. With this concept, the gust-induced wing loads can be significantly reduced in terms of root bending moment (RBM) by implementing the TWT. This twist changes the angle of attack (AoA) of the wingtip compared to the wing’s root, which is not actively controlled or adjusted by the pilot or aircraft systems during flight but occurs due to the wing’s structural and aerodynamic characteristics. When the shaft is positioned in front of the aerodynamic center, gust-induced aerodynamic forces cause the TWT to twist downward. It capitalizes on aerodynamic load alleviation from the TWT in conjunction with the wing’s aeroelastic effects. This approach minimizes the adverse impact of gusts on the airframe and enhances overall flight performance. The research indicates that there is a narrow range of optimal parameters for the TWT design, and the selected parameters show a reduction of 16% in the bending moment at the wing root due to gust [15].

The full-scale flying-wing aircraft (with half-span planform), including landing gears, engines, and fuel tanks as considered in wind tunnel study, is depicted in [16]. This aircraft exhibits tailless features, a high sweep angle, and an outer wing with a high aspect ratio, measuring 2 m in length for each wingtip’s PGAD. The aircraft’s structure is categorized into three sections: the inboard wing extending from the body’s center to the kink section, the outboard wing, and the PGAD. For the inboard wing, a multi-spar airframe layout was adopted, spanning 10.8 m from the body center to the wing kink section. In contrast, the outboard wing, with its high aspect ratio, follows a conventional two-spar layout, with the front and rear spars located at 15% and 75% of the local chord, respectively [16].

In a very recent work, a comparative study of folding wingtip (FWT) and PTWT was investigated for performance reduction in RBM and gust alleviation [17]. The conclusions reveal that TWT is superior in reducing RBM and gust alleviation when the hinge is placed as close to the leading edge as possible. However, the position of the hinge and spring stiffness is not well-proven in theory. Furthermore, the results show that the FWT performs better in terms of flutter reduction than the PTWT. This study has been validated by multiple works [18,19,20,21,22].

The lack of study of PTWT regarding its effects on static aeroelasticity (lift effectiveness, and divergence speed) and dynamic aeroelasticity (flutter speed) necessitates further investigation in this study. Aeroelastic phenomena have been extensively studied due to their critical impact on aircraft performance, particularly the interaction between aerodynamic forces and structural dynamics of aircraft [23]. These phenomena can lead to structural failure, which must be mitigated from the conceptual design phase to production design to ensure the aircraft operates safely within critical speed limit [8,9,10,24,25]. The primary aim of this research is to propose the guide mode preference technique and investigate the structural and aeroelastic implications of implementing a passive twisting wingtip. The research aims to understand how the integration of PTWT influences the aeroelastic and structural response of a wing.

All research methodologies are presented in Section 2, including the TWT model, twisting wingtip proving divergence speed, aeroelasticity, guide mode preference, optimization problem, aircraft wing model, and TLBO-DA optimizer. The design experiment results and discussions are presented in Section 3, while the conclusions are drawn in the final section, Section 4.

## 2. Research Methodology

The exploration of advanced wing technologies encompasses various innovative mechanisms aimed at enhancing aircraft performance and stability. One such mechanism is the TWT model, which plays a pivotal role in mitigating aerodynamic and aeroelastic phenomena. The TWT model integrates components such as the spar, rib, and twisting shaft (spring stiffness and shaft position) to facilitate controlled twisting motions at the wingtip. This controlled motion effectively adjusts the wing’s angle of attack, thus improving the aircraft’s stability and ride comfort during flight conditions. Additionally, the TWT mechanism is accompanied by an airfoil free body diagram (FBD), providing a comprehensive visualization of the forces and moments acting on the airfoil. Through detailed equations and derivations, the TWT model elucidates the relationship between aerodynamic parameters, structural elements, and dynamic responses, offering valuable insights into aircraft aeroelasticity and optimization problem.

### 2.1. Twisting Wingtip Model

The passive twist wingtip is a specialized component designed to address the challenges posed by gust-induced aerodynamic forces on aircraft wings. It comprises a distinct wing segment connected to the wingtip via a shaft and its equivalent stiffness spring, which are linked to the wing front spar to make it strong enough to sustain aerodynamic load [13,14,15,16]. When subjected to gusts, the aerodynamic forces cause the TWT to twist downward, resulting in a negative angle of attack (AoA) and, consequently, a negative aerodynamic force on the TWT. This twist helps alleviate the adverse effects of gusts on the wing, contributing to enhanced stability and performance during flight. Several key design parameters influence the effectiveness of the TWT. These include its dimensions, shaft position along the chord wise direction, torque spring stiffness, and the mass distribution and location of the center of gravity (CG). Careful consideration of these parameters is essential to ensure optimal performance and reliability of the TWT in various flight conditions. The TWT can be shown in Figure 1. To meet design requirements, the equivalent torque spring must possess sufficient stiffness to maintain the desired aerodynamic angle during level flight while limiting the maximum twist angle to prevent stalling under gust conditions. Present studies have provided valuable insights into the optimal twisting shaft geometry and the ideal location of the shaft for sustaining aeroelastic (lift-effectiveness, divergence speed, and flutter speed), and structural analysis (equivalent stress, buckling factor, mass, and mode shape). Additionally, finite element (FE) modeling is employed to analyze the structural behavior of the TWT and replace the original fixed wingtip.

Our main body wing with a TWT dimensional parameters design has been taken from the actual dimensions and specifications by using geometrical measurement to maintain the same aspect ratio when compared with the real model of MQ-1 predator [26], the corresponding parameters for the design are given in Table 1:

Here are the functions of the mentioned parts in the above model:(1)Spar: The spar is a structural component of the main body wing (MBW) that runs from the root (or fuselage attachment point) to the wingtip. Its primary function is to provide structural support and strength to the wing, helping it withstand the aerodynamic forces and loads experienced during flight. The spars of both TWT and MBW are in the same position.(2)Rib: Ribs are structural elements within the wing that are spaced along the spar. Their main function is to give the wing its airfoil shape and provide support for the wing covering or skin. Ribs play a crucial role in maintaining the wing’s structural integrity and aerodynamic performance.(3)Twisting Shaft: The twisting shaft is a component of the TWT system. Its function is to provide a controlled twisting motion to the TWT. This twisting motion changes the angle of attack of the wing, helping to reduce the aerodynamic forces induced by gusts, which affects the aerodynamic and aeroelastic characteristics, thus improving the aircraft’s stability and ride comfort during turbulent conditions.(4)TWT Hinge Line: The hinge line is the axis around which the PTWT at the wingtip can rotate. It serves as the pivot point for the TWT’s twisting motion. By strategically placing the hinge line in front of the wing’s aerodynamic center, the TWT can respond to gust loads by twisting the nose down, thereby alleviating the aerodynamic forces on the wing and improving its performance during flight.

### 2.2. Twisting Wingtip Proving Divergence Speed

In this section, the selection of the hinge position (HP) or the position of the shaft in the rod in MBW-TWT is crucial for our analysis. Placing the shaft at an optimal location helps provide a twisting motion, minimize stress concentrations, and ensures that the wing can withstand aerodynamic forces and other external factors without experiencing excessive deformation or failure. Additionally, another important consideration is longitudinal stability. Optimal shaft positioning allows for precise control over pitch changes, ensuring that the aircraft responds and maintains longitudinal stability.

Placing the shaft at the front of aerodynamic center (AC) can significantly affect the aerodynamic characteristics of the wing. The AC is the point along the chord where the lift force can be considered to act. When the shaft is positioned at the AC, any changes in wing twist induced by the main body wing-twisting wingtip system directly affect the lift force without introducing additional moments. This configuration may lead to instability issues, particularly if the control system is not properly designed to handle the dynamics. Small perturbations in the lift force can quickly amplify, leading to divergence in speed if not properly controlled. Placing the shaft at a longer distance, such as 0.25c (25% chord) from the leading edge, provides greater leverage for controlling wing twist. This would give the moment due to the lift force in the opposite direction to the moment due to the aircraft’s weight. The increased leverage can amplify control inputs or external disturbances, potentially leading to divergence in speed if the system is unable to adequately dampen oscillations. The following derivation presents the appropriate hinge position to counteract divergence speed.

An airfoil free body diagram (FBD) with a twisting wingtip mechanism can help illustrate the forces and moments acting on the airfoil. Let us explain and define each component in the context of the given FBD (Figure 2):(1)Center of gravity (CG): The CG represents the center of mass of the TWT. It is the point around which the wing’s weight is balanced. The location of the CG is crucial for overall stability and balance during flight.(2)Weight (W): The force acting vertically downward due to the gravitational attraction between the TWT and the earth. It represents the total mass of the TWT times the acceleration due to gravity (W = mg).(3)Lift (L): The aerodynamic force generated by the airfoil. It acts perpendicular to the relative airflow and opposes the weight of the TWT.(4)Chord length (c): The distance typically represents the chord length of the airfoil. The chord is the straight-line distance between the leading edge and the trailing edge of the airfoil.(5)Hinge position (e): The distance is often used to denote the distance between the center of aerodynamic and the elastic axis or hinge position (HP) of the wing, but for this case it measures from HP or twisting shaft position to aerodynamic center.(6)The CG position (a): The distance is denoted as the center of gravity (CG) and the hinge position. The aerodynamic center is the point on the airfoil’s chord line where changes in angle of attack (α) do not affect the aircraft’s pitching moment.(7)Speed (V): The relative velocity of the airfoil through the air. It plays a significant role in determining the magnitude of the lift force and other aerodynamic effects.

From the above model, we have defined the angle θ as follows:(1)$$\theta =\theta_{0}+\theta_{e}$$

From Equation (1), $\theta ,\;\theta_{0},$ and $\theta_{e}$ are the actual twist angle, baseline twist angle and extra twist angle, respectively. Take a moment of all forces around the HP to obtain the expression for the extra twist angle applied to the wingtip yields (clockwise is positive):(2)$$\sum M_{HP}=M_{AC}-Le-k_{\theta }\theta_{e}$$(3)$$L=q{sC}_{L}$$(4)$$M_{AC}=qse\;C_{MAC}$$(5)$$C_{L}=C_{L0}+\frac{\partial C_{L}}{\partial \theta }\theta$$(6)$$C_{MAC}=C_{MAC0}$$

After solving for Equation (2), we obtain(7)$$\theta_{e}=\frac{qs}{k_{\theta }}\frac{\left(-e\frac{\partial C_{L}}{\partial \theta }\theta_{0}\right)}{1+q\frac{Se}{k_{\theta }}\frac{\partial C_{L}}{\partial \theta }}$$
where *s* represents the wing area of an aircraft. *q* stands for dynamic pressure, $q=\frac{1}{2}\rho v^{2}$. *C_L_* represents the lift coefficient. *k_θ_* represents equivalent angular stiffness of the twisting shaft, which in the context of aircraft or structural mechanics, refers to the stiffness of an element or component with respect to angular deformation (rotation or twisting). The angle $\theta_{e}$ can increase to infinity if the division term approaches zero, but in this case, it never occurs due to the position of the hinge placed at the front of the aerodynamic center. From the derivation, it can be proven that the position of twisting wing shaft should be placed at the front of aerodynamic center, ensuring that the divergence speed never damages the TWT. The aerodynamic center is at a quarter of chord length. The exact position of HP needs to be studied in our research.

### 2.3. Aeroelasticity

Aeroelasticity is the physics that deals with the mutual interaction between inertia, elastic, and aerodynamic forces [1]. It is usually found to be the cause of structural failures and performance reduction during flight. Aeroelasticity can be classified as static or dynamic aeroelastic phenomena. Static aeroelastic parameters that are usually included in the design optimization of a wing structure are lift effectiveness and divergence, while the most significant dynamic aeroelastic design parameter is flutter. Flutter is the dynamic aeroelastic instability of an aircraft structure and consists of violent unstable oscillations. It is thought of as the critical speed that can be avoided in an aircraft design process. Similarly, divergence is regarded as the critical speed at which aerodynamic forces overcome structural restoration. Lift effectiveness, on the other hand, is the ratio of total lift at a particular wind speed when flexibility is considered in the total lift compared to when the wing is rigid. These parameters are important since a wing with TWT light weight and high lift effectiveness is desirable. Important parameters related to aeroelastic characteristics are aerodynamic and finite element analysis (FEA), and they are also considered in this section. These parameters can be computed by using FEA and aerodynamic panel methods as presented in [8,9].

#### 2.3.1. Structural Model

The parameters that need to be studied in static analysis are strain and stress while in dynamic analysis, natural frequency and mode shape need to be studied.

The dynamics equation for aircraft structure can be derived from second law of motion:(8)$$\left[M]\ddot{u}+[C]\dot{u}+[K]u=F(t\right)$$
where [M], [C], and [K] are the mass, viscous damping and stiffness matrix of a structure, respectively. u is the nodal displacement vector, and **F**(t) is the vector of external forces.

#### 2.3.2. Modal Analysis

The solution for free vibration analysis of the system in Equation (8) can be computed when external force, and damping term are neglected. The objective is to solve natural frequencies and mode shapes of the structure. If the structure has oscillation motion, it has(9)$$Z(t)=\bar{Z}\exp(\lambda t)$$

Substituting (9) into (8) gives(10)$$(A-\lambda_{i}I){\bar{Z}}_{i}=0$$
where **λ_i_** is an eigenvalue of the vibration system and ${\bar{Z}}_{i}$ is an eigenvector corresponding to **λ_i_**. For an *n* degree of freedom system, there are *m* eigenvalues and eigenvectors.

Given that $\left[z_{m}]=[{\bar{z}}_{1},\ldots ,{\bar{z}}_{m}\right]$ is the modal matrix containing the first *m* mode shapes of the structural system, by substituting $u=[z_{m}]x$ into Equation (8) and pre-multiplying by ${\left[z_{m}\right]}^{T}$, the reduced-order structural model can be obtained:(11)[Mg]x¨+[Cg]x˙+[Kg]x=F(t)g
where $\left[M_{g}]={\left[z_{m}\right]}^{T}[M][z_{m}\right]$, $\left[C_{g}]={\left[z_{m}\right]}^{T}[C][z_{m}\right]$, $\left[K_{g}]={\left[z_{m}\right]}^{T}[K][z_{m}\right]$, and $F_{g}={\left[z_{m}\right]}^{T}F$.

Equation (11) can be altered to become a discrete-time state space model as(12)$$[DEOM_{2}]r^{n+1}+[DEOM_{1}]r^{n}+F^{n+\frac{1}{2}}=0$$
where$$[DEOM_{2}]=\left[\begin{array}{cc} \frac{M_{g}}{\Delta t} & \frac{C_{g}}{\Delta t}+\frac{K_{g}}{2}\\ \frac{-M_{g}}{2} & \frac{M_{g}}{\Delta t} \end{array}\right],\;[DEOM_{1}]=\left[\begin{array}{cc} \frac{-M_{g}}{\Delta t} & \frac{-C_{g}}{\Delta t}+\frac{K_{g}}{2}\\ \frac{-M_{g}}{2} & \frac{-M_{g}}{\Delta t} \end{array}\right]F^{n+\frac{1}{2}}=\left\{\begin{array}{c} -F_{g}\\ 0 \end{array}\right\}$$

#### 2.3.3. Aerodynamic Model

The model of aerodynamic load on the aircraft wing assumes subsonic flow. The fluid dynamic flow equation is reduced to the Laplace equation and irrotational flow. The aerodynamic loads on a wing can be computed by using the vortex ring method [8]. The aerodynamic model is based on the unsteady vortex lattice method (UVLM), with is sufficiently accurate for analyzing the TWT model. This technique is known for its computational efficiency in aeroelastic analysis at low speed. Due to the high computational cost of aeroelastic analysis and optimization, this method is adopted in this study. Using this approach, the main body wing and twisting wingtip (MBW-TWT) are modeled as lifting surface without thickness. The lifting surfaces are then discretized into multiple panels for vortex lattice method (VLM) analysis, as shown in Figure 3a. The TWT is connected to the MBW at the same sweep angle (Λ) through a twisting shaft with a small gap, allowing for smaller panel modeling. The twisted configuration of the TWT is illustrated in Figure 3b.

Based on Biot–Savart law, the aerodynamic model is described as(13)[AIC]Γ={RHS}
where [*AIC*] is the aerodynamic influence coefficient, {*RHS*} is a right-hand side vector and Γ is a vortex strength. This equation can be converted to a discrete-time equation as(14)$$[CDR_{2}]\Gamma^{n+1}+[CDR_{1}]\Gamma^{n}=W^{n+1}$$
where $W^{n+1}$ is the downwash vector. Consequently, the pressure difference between the upper and the lower surface of the panels can be determined using the relationship.(15)$$\Delta P^{n+\frac{1}{2}}=[C2P_{2}]\Gamma^{n+1}+[C2P_{1}]\Gamma^{n}$$

Interface between structural and aerodynamic forces is carried out by means of surface spline interpolation [27] because the downwash can be changed to(16)$$W^{n+1}=U_{\infty }[H]u=U_{\infty }[H][Z_{m}]x=[WDR]r$$
where [H] is a transformation matrix and $U_{\infty }$ is a free stream velocity. Combining Equations (14)–(16), the aerodynamic forces can be transformed to structural nodal forces by(17)$$F^{n+\frac{1}{2}}=[CNFR_{2}]\Gamma^{n+1}+[CNFR_{1}]\Gamma^{n}$$
where [*CNFR*_2_] and [*CNFR*_3_] are force transformation matrices.

#### 2.3.4. Flutter Speed

A discrete-time aeroelastic analysis of a wing structure can be achieved by combining Equations (12), (14), (16) and (17) leading to(18)$$\left[\begin{array}{cc} \left[CDR_{2}\right] & -[WDR]\\ \left[CNFR_{2}\right] & \left[DEOM_{2}\right] \end{array}\right]{\left\{\begin{array}{c} \Gamma \\ r \end{array}\right\}}^{n+1}+\left[\begin{array}{cc} \left[CDR_{1}\right] & \left[0\right]\\ \left[CNFR_{1}\right] & \left[DEOM_{1}\right] \end{array}\right]{\left\{\begin{array}{c} \Gamma \\ r \end{array}\right\}}^{n}=0$$

Flutter analysis is carried out in such a way that, at a particular wind speed, the eigensystem (18) is obtained, and the eigenvalues can be computed. The speed at which one of the real parts of the continuous-time eigenvalues starts to become positive is taken as the flutter speed (*V_f_*) [8].

#### 2.3.5. Lift Effectiveness

Lift effectiveness is the ability of the wing to produce lift force when it has a lift incidence. The lift effectiveness is the ratio of total lift force on a flexible wing to that of its rigid counterpart.(19)$$\eta_{L}=\frac{L_{F}}{L_{R}}$$
where $L_{F}=qS_{a}{\left[AIC\right]}_{F}\alpha$ is the total lift force due to flexible wing, $L_{R}=qS_{a}{\left[AIC\right]}_{R}\alpha$ is the total lift force considering a wing being rigid, $S_{a}$ is the diagonal matrix of panel areas, *q* = $\frac{1}{2}\rho_{air}U_{\infty }^{2}$ is the dynamic pressure, **α** is the vector of panel angles of attack, and [*AIC*] is the aerodynamic influence coefficient.

#### 2.3.6. Divergence Speed

Divergence speed (*V_d_*) can be computed by solving the eigenvalue problem.(20)$$\left[[SIC]S_{a}{\left[AIC\right]}_{R}-\lambda_{D}I\right]\Delta \alpha =0$$
where *λ_d_* = 1/*q* is an eigenvalue, [*SIC*] is the structural influence coefficient matrix, and Δ**α** is the vector of deformation angles. The largest real positive eigenvalue of Equation (20) is the wing divergence speed. The divergence speed formula in this section is used to calculate the divergence speed and confirm that it never occurs of *V_d_* in Equation (7) if e is positive as mentioned previously.

### 2.4. Structural Analysis

#### 2.4.1. Stress Analysis

The finite element technique divides the aircraft structure into small pieces or finite elements to perform global stiffness **K**. For solving static analysis in (8), the applied force on the structure is derived by aerodynamics analysis $L_{R}=qS_{a}{\left[AIC\right]}_{R}\alpha$ as mentioned in the previous part. The aerodynamics load supplies to aircraft structure (8), and the displacement vector can be solved. For linear analysis of the structure, Hooke’s law can represent the relation of strain and stress. The stresses inside the aircraft structure can be solved. In this study, the equivalent stress in form of Von Mises equivalent stress is expressed as follows:(21)$$\sigma ={\left(\sigma_{11}^{2}+\sigma_{22}^{2}-\sigma_{11}\sigma_{22}+3\sigma_{12}^{2}\right)}^{1/2}$$
where $\sigma_{ij}$ is the stress component.

#### 2.4.2. Buckling Factor

The buckling factor must be considered in the design constraints of the aircraft wing structure to avoid structural failure. Aircraft wing structures are usually considered as shell structures, so buckling analysis is a static analysis due to in-plane loads of shell structures. The buckling analysis will be explained as follows and can be used for solving global buckling of the structures.

By taking the strain energy of aircraft structure and the work due to bending of the plate in the structure, and using the energy method, the static system for solving buckling is(22)$$\left[K+\lambda G\right]\;u=0$$
where **K** is the stiffness matrix, **G** is called the geometrical stiffness matrix, and λ is the ratio of critical loads to the applied loads.

Since **u** is arbitrary, the system is an eigenvalue problem with *n* eigenvalues that can be either positive or negative. The eigenvalues determine the critical loads (normally taking the minimum eigenvalue is enough for structural constraints), while eigenvectors of the problem give the buckling shape corresponding to their eigenvalues.

### 2.5. Flow Diagram for MBW-TWT Structural Design and Aeroelastic Analysis

The process for optimization and function evaluation can be set up and arranged in a flow of steps as shown in Figure 4.

### 2.6. Design Experiment

#### 2.6.1. Guide Mode Preference

To find the optimized mass for our wing system model that compared with the MQ1 aircraft model and the inequalities of constraints can be shown in the optimization problem solving model on the single objective function *f* below:(23)$$\min\;\;f(x)=w_{1}M+w_{2}(\omega_{ng}-\omega_{nj}),j=1,\ldots ,5$$
where *f* is the objective function that shows the weighted sum of wing mass and the difference between the modal mode frequency and the guide mode frequency. ***x*** is the design variable vector composes of the radius of the shaft (*r*), shaft position, and structural thickness (*x_i_*). The total number of design variables is 546. The weighing factors for those are *w*_1_ = *w*_2_ = 0.5. *M* is the optimized mass of MQ1 aircraft model. $\omega_{ng}$ is the natural frequency or the guide mode preference when the TWT rotates with a simple rotational mode. The guide mode is used to suppress the first one mode of the overall MBW-TWT relying on twisting mode, which is main function of TWT. The guide mode mainly relies on shaft stiffness (geometry of shaft), and TWT’s moment of inertia (geometry of TWT). $\omega_{nj}$ is the 1st–5th modal frequencies, where *j* = 1, 2, …, 5.

#### 2.6.2. Optimization Problem

The single objective function is subject to the following inequalities constraints:*σ* ≤ *σ*_y_: The maximum stress, *σ,* on the system must not exceed the yield stress *σ*_y_ to maintain all structures of MBW-TWT as safe.*λ* ≥ 1: The buckling factor is larger or equal to 1 to maintain all thin structures as not failing by buckle.*shfpos* ≤ 0.25c: The shaft position must not exceed the aerodynamic center of the airfoil to maintain TWT structure, which can restore aerodynamic force without fail as derived in (7).0.9*t_i_* ≤ *r* ≤ 0.95*t_i_*, *i* = 1, …, 4: The radius of the shaft cross section must be bounded.*V_d_* ≥ 100 m/s: The divergence speed must be larger, as much as possible.*V_f_* ≥ 60 m/s: The flutter speed must also be larger than 60 m/s.

The optimization problem can be seen in the following equation:(24)$$min\;f=w_{1}M+w_{2}(\omega_{ng}-\omega_{nj}),j=1,\ldots ,5$$subject to   *w*_1_ = *w*_2_ = 0.5*σ* ≤ *σ*_y_*λ* ≥ 1*Shfpos ≤* 0.25c0.9*t_i_* ≤ *r* ≤ 0.95*t_i_*, *i =* 1, …, 4*η_L_* ≥ 0.95*Vd* ≥ 100 m/s*Vf* ≥ 60 m/s0.005 ≤ *x_i_* ≤ 0.020 m

### 2.7. Aircraft Wing Model

In this research, MBW and TWT are modeled from the MQ-1 Predator as detailed in Table 1. The structural model composed of MBW and TWT is shown in Figure 5, while the internal structures, including skin variables, are presented in Figure 6. The present research selected Aluminum alloy 7075 as the material for the whole part, except for the twisting shaft made from steel. The material properties of the wing and twisting shaft are shown in Table 2.

The TWT structure with its twisting shaft (red solid line) comes from MATLAB, while the Ansys model can be shown in Figure 7. All structures of MBW and TWT are modeled with shell63, while the twisting shaft is modeled with beam188 with reasons that can be seen in our previous research [10]. The twisting shaft position is discretized into four sections according to the rib section, as shown in Figure 7a.

### 2.8. TLBO-DA

The design problems will be solved by the recently updated optimizer, called the teaching–learning-based optimization with a diversity archive (ATLBO-DA). The algorithm has already proven its effectiveness in dimensional synthesis in previous four-bar mechanism research and was recently updated for work in [28]. The variables of the optimizer are determined as IReset = 20, IRange = 5, and *δ* = 1. The population is set up as $n_{p}$ = 30, while the maximum iteration is 100. The weight is set at 0.5, as studied in previous work [28].

## 3. Design Results and Discussions

The optimum results after solving the optimization problems with objectives and constraints will be shown in Table 3. The optimum result is validated by identifying the elastic axis of the MBW-TWT. The well-designed MBW should provide strong structural support for the hinge or twisting shaft of the TWT, which should be positioned as close to the aerodynamic center as possible, as shown in Figure 8. The technique used to determine the elastic axis is adapted from [8] through finite element analysis (FEA), where a torque is applied at the wingtip, and radial-basis function interpolation is used to find the locus of wing sections with zero deflection. A line passing through all points along the span is used to approximate the elastic axis of the wing.

The elastic axis is located along the wing at approximately 0.4637 m from the leading edge and trends toward the aerodynamic axis at the wingtip. In theory, the TWT hinge position should not be placed beyond the aerodynamics axis to prevent divergence speed failure. However, if the elastic axis at the tip of the MBW is closely aligned with aerodynamics axis, it supports the theoretical model and simplifies handling of the TWT device.

The results are consistent with [8], where the highest lift-effectiveness is achieved when the elastic axis is positioned as far as possible from the aerodynamic axis. Conversely, in this design, the elastic axis is positioned closer to the aerodynamics axis near the wingtip to facilitate TWT operation. This placement results in lower lift-effectiveness efficiency. More details on lift-effectiveness are discussed in the next paragraph.

Since the MTOW of MQ-1 is 1020 kg, the mass ratio of wing to MTOW is 0.435 (43.5%). In general, for most aircraft, including commercial airliners and general aviation planes, the wings contribute approximately 20–30% of the MTOW. Therefore, our wing design is unconventional, which necessitates the MQ1 to have a stronger structure to handle TWT system. The stronger structure can handle lower maximum stress, higher buckling factor, and moderate displacement. The buckling factor is close to the buckling design constraint to maintain lift-effectiveness, as mentioned in [8]. The design results satisfy the constraints for both static aeroelasticity (lift-effectiveness and divergence speed) and dynamic aeroelasticity (flutter). Lift-effectiveness is a critical parameter in morphing aircraft design, as it represents the contribution of the TWT to the main body wing. The lift-effectiveness from this design is slightly higher than the threshold limit state function. To maintain structural integrity while ensuring optimal TWT performance, the lift-effectiveness should be constrained. Any improvement in lift should be achieved through active TWT adjustments during operation, which will be studied in the next paragraph. Flutter is a critical speed that must be avoided in aircraft design, as noted in [9]. The results confirm that the design effectively prevents flutter within the specified constraint. Although a stronger structural design cannot eliminate flutter entirely, it can mitigate its effects by modifying the mode shapes. Especially for the divergence speed value, it confirms the proposed basis in the design of the twisting shaft position of TWT, which must not be placed over the aerodynamic center at 0.2103c from the leading-edge with equivalent stiffness 77,655 Nm/rad to counteract the divergence speed and aerodynamic load. This position has been referenced in all previous works [13,14,15,16,17] without explicit proof. Additionally, the displacement exceeds the allowable displacement of the wing to enable the TWT to twist freely around the twisting shaft.

The design results for the modal frequencies for 1st to 4th modes of the aircraft wing can be obtained in the following ANSYS results as shown in Figure 9a–d.

The corresponding modal frequencies that are obtained from the ANSYS results can be summarized in Table 4.

The results verified that the first mode and the second mode are torsional modes. This result is derived from the proposed guide mode preference technique in this study. Fortunately, these modes persist in torsional mode, providing confirmation for steady torsional stress on the rod over the longer time interval, ensuring that the state of stress of the rod does not change abruptly or prematurely. In general, flutter occurs due to the coupling of bending and torsional modes. The guide mode preference technique shifts the bending mode to the third mode, ensuring that the flutter speed increases to comply with the design constraints.

Knowing the twist angle values of a TWT provides critical insights for aircraft design and performance. Firstly, it directly impacts aerodynamic force by changing the lift distribution along the wingspan. A proper twist can increase/reduce lift effectiveness, enhancing fuel efficiency if the TWT is designed to generate sufficient lift force. Secondly, twist angles play a crucial role in stall prevention and control. Negative twist (washout) delays stall onset at the wingtip, allowing pilots to maintain control during critical flight conditions. Additionally, twisting affects aeroelastic behavior, such as flutter and divergence, as detailed in the following results. The effect of twist angles of TWT (*θ*) on aeroelastic phenomena is present in Table 5. Graphical representations of twisting angle versus aeroelastic phenomena are presented in Figure 10, Figure 11 and Figure 12. The results indicate that an increase in the TWT angle (*θ*) leads to a slight increase in lift-effectiveness, whereas it decreases in negative sign as shown in Table 5 (column 2) and Figure 10. The percentage change in lift efficiency is calculated using relative error statistics, with the highest efficiency observed at approximately 0.68% at a twisting angle of 30°.

This result suggests that incorporating the TWT does not significantly impact the main body wing (MBW), making it feasible to integrate the TWT as a retrofit device without requiring major structural modifications. This lift-effectiveness trend aligns with the previous research findings [8,9,10]. The relationship follows an exponential curve, reflecting how adjustment of the TWT directly impacts aerodynamic changes and affects control law and stability of aircraft. Lift-effectiveness can be increased by scaling up the TWT, with the overall trend remaining consistent with the findings in this study. Divergence speeds can potentially damage the wing structure if aerodynamic forces overcome structural restoration. The theory proposed in Equation (7) suggests that the twisting shaft position should not be placed beyond the aerodynamic center to prevent failure due to divergence speed and to maintain stability. The results align with previous works [13,14,15,16,17], particularly the conclusion in [17] that the twisting shaft should be positioned close to the leading edge. The optimal twisting shaft location is at 0.2103c and the divergence speed confirms the proposed theory. Adjusting the TWT angle can increase the divergence speed in all cases because it directly affects the torsion stiffness of the twisting shaft. This torsion transfers to twist the MBW, altering the global stiffness by pre-stress, as demonstrated in previous studies [8,9]. The trim angle of the MBW-TWT is calculated using the same technique adapted from [8] through the vortex lattice method and the bisection method, yielding a trim angle of 8°. If the TWT operates at the first torsional mode at the trim angle, the maximum twisting angle is 23.388°, meaning that the possible divergence speed does not reach 25°. The results indicate that the highest divergence speed occurs at both *θ* = ±25°. Within the interval of ±20°, divergence speed variations remain minimal, as shown in Table 5 (column 2) and Figure 11, which is consistent with previous research findings [8,9]. The highest efficiency of this device for improving divergence speed is 27.7% at 20° (calculated using relative error statistic), as illustrated in Figure 11. The flutter speed of the wing exhibits a unique trend as the TWT angle changes, with a positive but limited effect, as shown in Table 5 (column 3) and Figure 12. The twisting of the TWT slightly increases flutter speed due to the guide mode preference technique, which suppresses the first mode and second modes, ensuring that only torsional mode is present while shifting the bending mode to the third mode. The change in TWT angles does not impact mode shape transformation. The optimized TWT design helps maintain flutter speed at a safe margin away from the flight envelope. However, the TWT is less effective in increasing flutter speed compared to the folding wingtip [17]. The relative percentage change in flutter confirms this observation, with the highest improvement recorded at 1.84% at the maximum twisting angle, as shown in Figure 12.

## 4. Conclusions

This research presents a conceptual design of TWT integrating a twisting shaft, which can be designed to pre-modes such as torsional mode by utilizing a technique of guide mode preference. The proposed wingtip structure design can enhance the aerodynamic performance and control of unmanned aerial vehicles (UAVs) under acceptable constraints such as lift effectiveness, divergence speed, and flutter speed. The simplified design and numerical experiment can reveal the optimum structure for use in the TWT, including the twisting shaft position and radius, twisting mode, and aeroelastic properties. Especially, the theory proposed to prevent TWT failure by divergence has been validated through numerical experiments. The numerical results demonstrate that the TWT performance can enhance both the dynamic and static aeroelasticity of the original wing, rather than solely focusing on flutter, and gust alleviation. The divergence speed results confirm the twisting shaft position should not extend beyond the aerodynamic center at 0.2103 of the chord length, in accordance with the proposed theory. The adjustment of TWT angle validates that the highest efficiency of divergence speed is in the range of 0–27.7%. Additionally, the relative aeroelastic efficiencies indicate that the maximum lift-effectiveness is 0.68% at the twisting angle 30° in both upward and downward directions, following an exponential relationship. This relationship can be extended to aircraft control law if it has adequate design to generate more additional lift. However, the relative efficiency of flutter speed enhancement is limited when using the TWT, with improvements ranging from 0 to 1.84% when twisting up and down, aligning with finding from previous research.

In future work, the proposed conceptual design of TWT can be extended to characterize gust alleviations with reliability-based design.

## Figures and Tables

**Figure 1 biomimetics-10-00110-f001:**
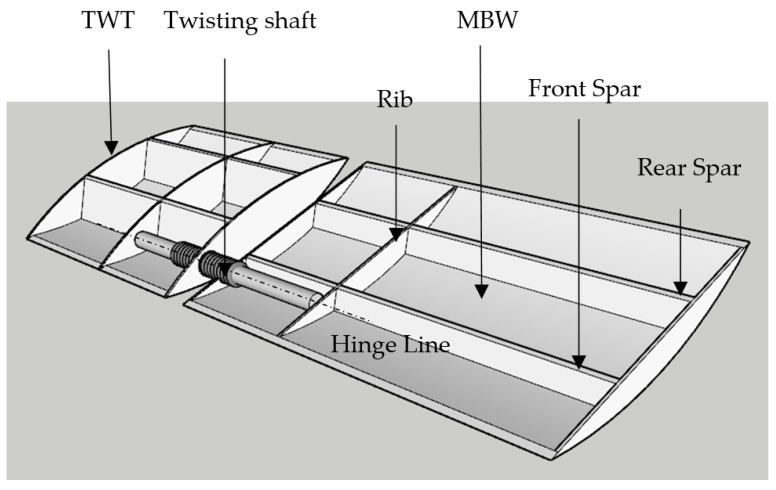
The model of TWT and main body wing (MBW).

**Figure 2 biomimetics-10-00110-f002:**
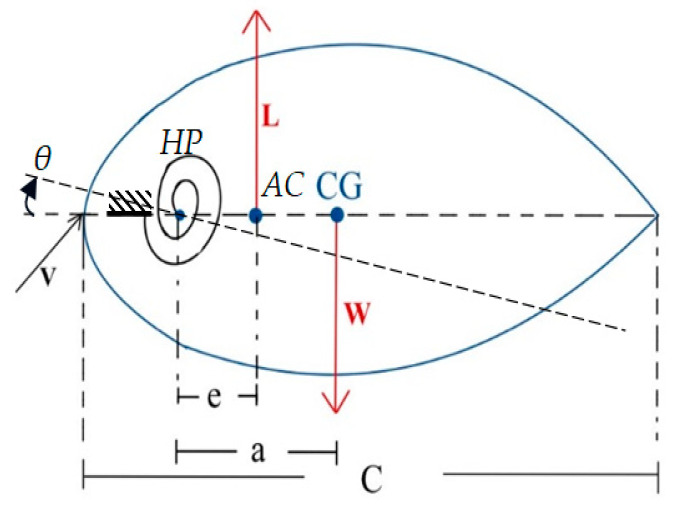
Free body diagram of an airfoil.

**Figure 3 biomimetics-10-00110-f003:**
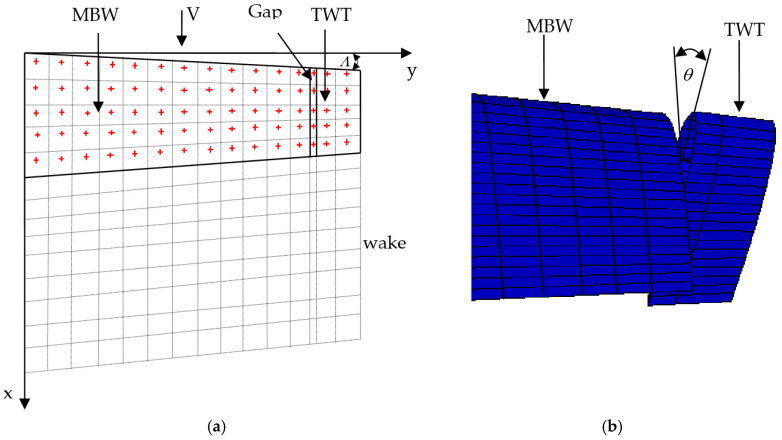
(**a**) MBW-TWT aerodynamic model (+ collocation point). (**b**) Twisted TWT.

**Figure 4 biomimetics-10-00110-f004:**
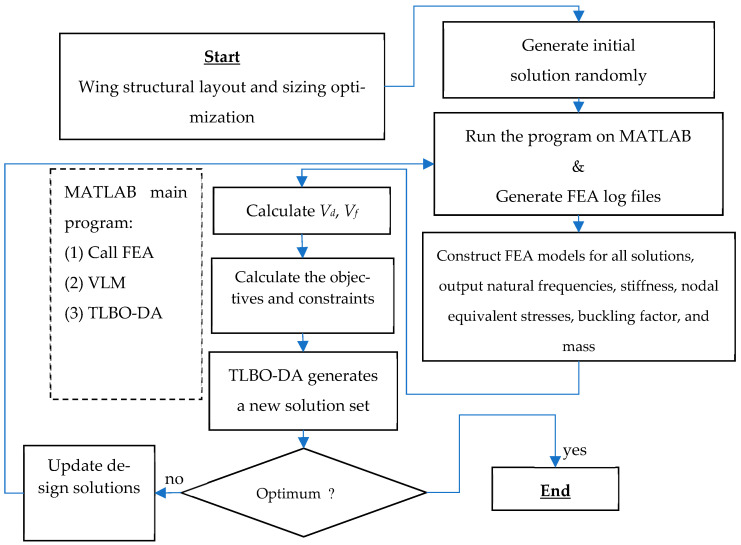
Process for optimization and function evaluation diagram.

**Figure 5 biomimetics-10-00110-f005:**
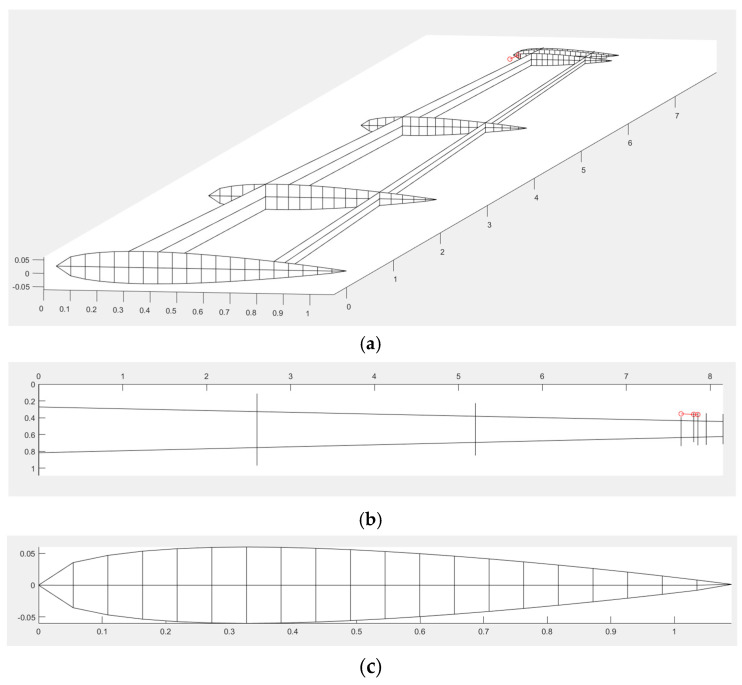
(**a**) The three-dimensional view of the main wing with the twisting wingtip and twisting shaft (-o-). (**b**) The top view of the main wing with the twisting wingtip and twisting shaft (-o-). (**c**) The cross-section view of the main wing that shows the airfoil of the wing root.

**Figure 6 biomimetics-10-00110-f006:**
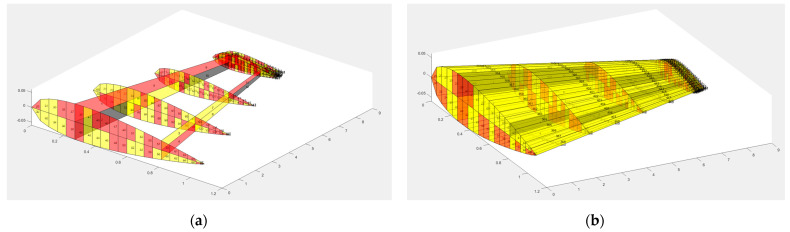
All design variable thickness of MBW and TWT: (**a**) internal structure design variables; (**b**) internal structure included skin variables.

**Figure 7 biomimetics-10-00110-f007:**
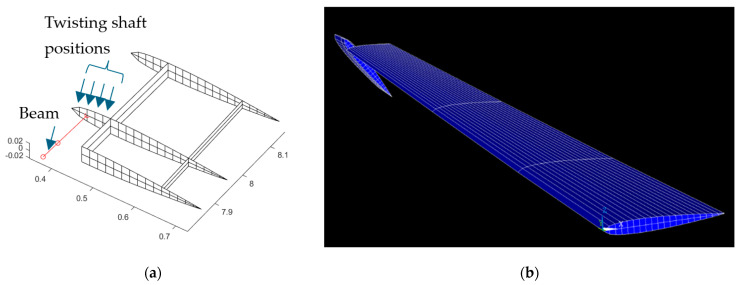
(**a**) The TWT structure installed shaft rod that is shown as the red solid line. (**b**) Ansys model of the whole aircraft wing.

**Figure 8 biomimetics-10-00110-f008:**
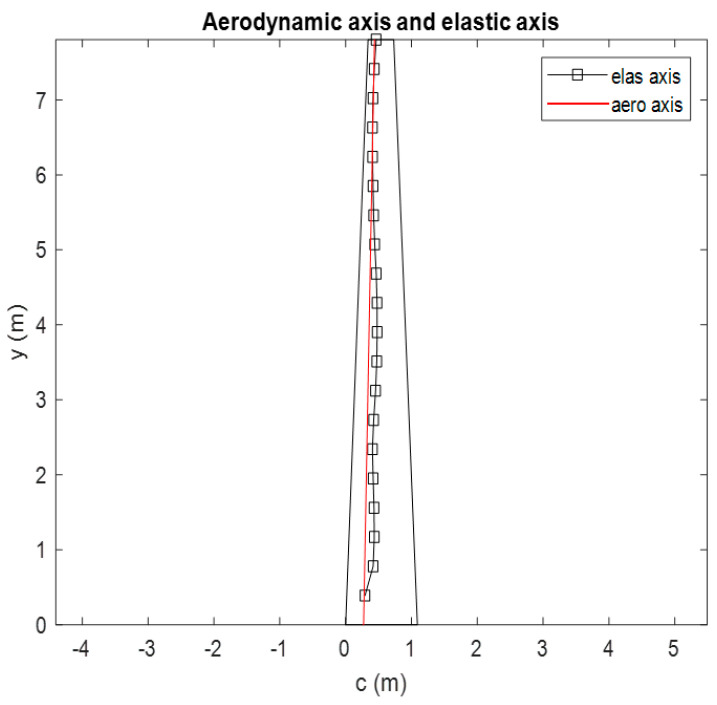
Aerodynamic axis and elastic axis of the optimum design result.

**Figure 9 biomimetics-10-00110-f009:**
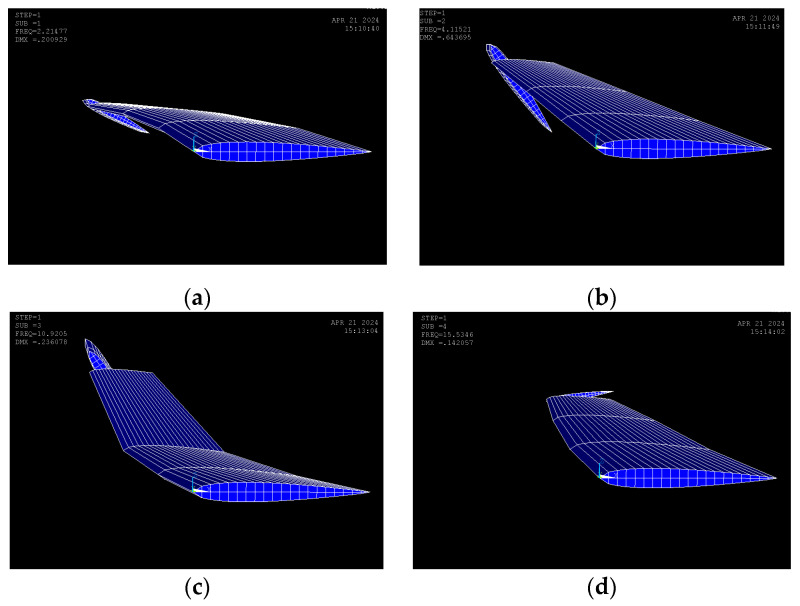
Aircraft wing modal frequencies: (**a**) 1st mode torsion; (**b**) 2nd mode torsion; (**c**) 3rd mode bending; (**d**) 4th mode sway.

**Figure 10 biomimetics-10-00110-f010:**
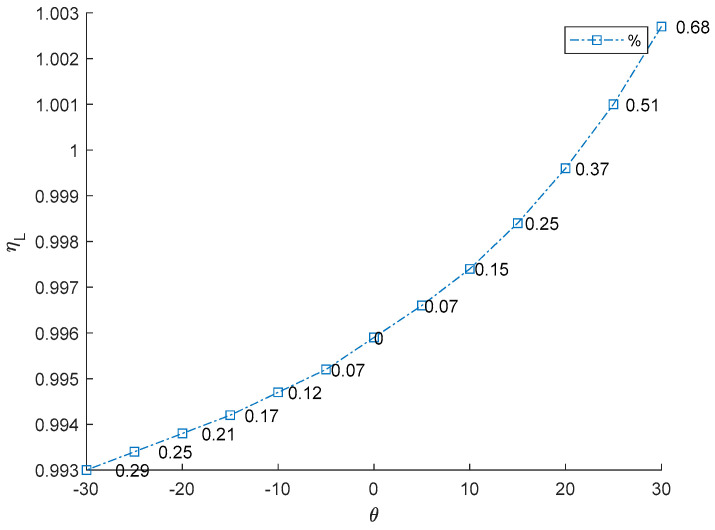
Relationship between TWT angle and lift effectiveness.

**Figure 11 biomimetics-10-00110-f011:**
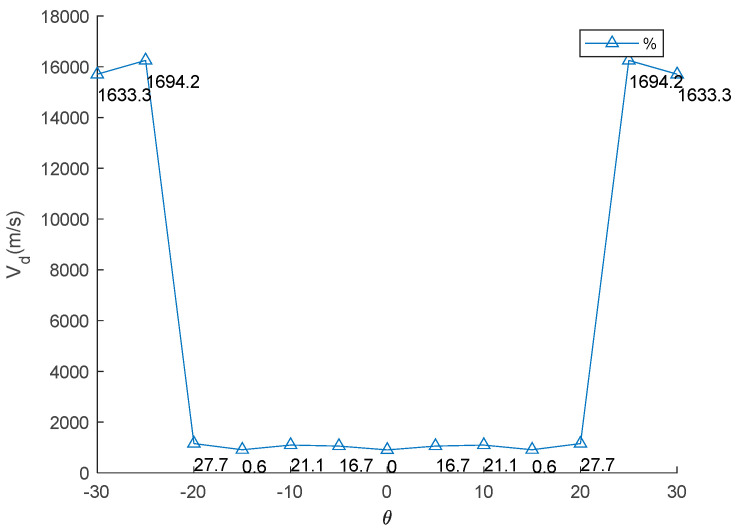
Relationship between TWT angle and divergence speed.

**Figure 12 biomimetics-10-00110-f012:**
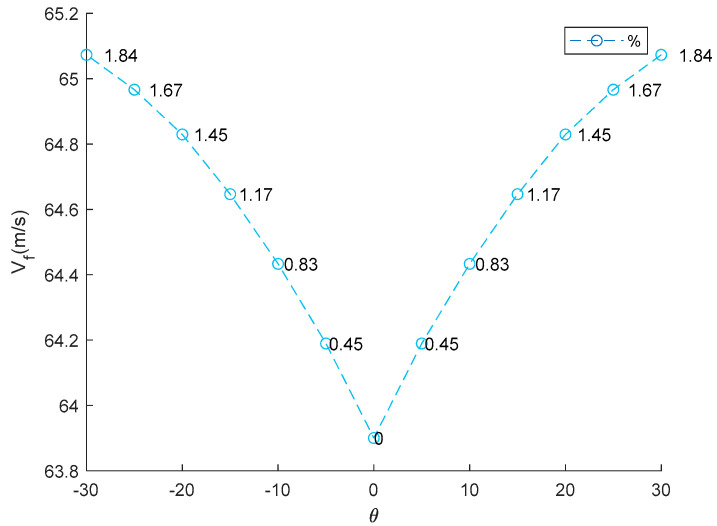
Relationship between TWT angle and flutter speed.

**Table 1 biomimetics-10-00110-t001:** MBW and TWT parameters.

No.	Parameters	Values	
		MBW	TWT
1	Semi-span length, L (m)	7.802	0.3
2	Root chord length, RC (m)	1.09	0.3858
3	Tip chord length, TC (m)	0.39	0.3589
4	Sweep angle Λ, (degree)	2.5°	2.5°
5	Number of ribs	4	3
6	Number of spars	2	2
7	Front spar position	25%	25%
8	Rear spar position	75%	75%
9	Gap	-	0.05
10	NACA	0012	0012

**Table 2 biomimetics-10-00110-t002:** Material properties.

Aluminum **Properties**	**Value**	**Unit**
Young’s modulus (E)	70 × 10^9^	Pa
Poisson’s ratio (ν)	0.3	-
Density (*ρ*)	2700	kg/m^3^
Steel **Properties**	**Value**	**Unit**
Young’s modulus (E)	207 × 10^9^	Pa
Poisson’s ratio (ν)	0.27	-
Density (*ρ*)	7700	kg/m^3^

**Table 3 biomimetics-10-00110-t003:** The optimum results of MBW and TWT.

Mass (kg)	1st Torsion Mode (rad/s)	Max Stress (MPa)	Buckling Factor	Displacement (m)	Lift-Effectiveness	Flutter Speed (m/s)	Divergence Speed (m/s)
443.9761	13.9158	21.724	1.2036	0.3277	0.9952	63.9004	905.9394

**Table 4 biomimetics-10-00110-t004:** The 5 modal frequencies of MBW-TWT.

1st Torsion Mode (Hz)	2nd Torsion Mode (Hz)	3rd Bending Mode (Hz)	4th Sway Mode (Hz)	5th Torsion and Bending Mode (Hz)
2.21477	4.11521	10.92054	15.53463	27.94027

**Table 5 biomimetics-10-00110-t005:** The effect of twist angles on aeroelastic phenomena.

TWT Angle	Lift Effectiveness	Divergence Speed (m/s)	Flutter Speed (m/s)
+30	1.0027	1.5703 × 10^4^	65.0730
+25	1.0010	1.6254 × 10^4^	64.9664
+20	0.9996	1.1565 × 10^3^	64.8293
+15	0.9984	911.0984	64.6466
+10	0.9974	1.0967 × 10^3^	64.4334
+5	0.9966	1.0573 × 10^3^	64.1897
0	0.9959	905.9394	63.9004
−5	0.9952	1.0573 × 10^3^	64.1897
−10	0.9947	1.0967 × 10^3^	64.4334
−15	0.9942	911.0984	64.6466
−20	0.9938	1.1565 × 10^3^	64.8293
−25	0.9934	1.6254 × 10^4^	64.9664
−30	0.9930	1.5703 × 10^4^	65.0730

## Data Availability

Data are contained within the article.

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
