# Peer review of "A New Conceptual Design of Twisting Morphing Wing"

_biomimetics, 2025, doi:10.3390/biomimetics10020110_

Round 1
Reviewer 1 Report
Comments and Suggestions for Authors
1. The proposed concept of a passive twisting wingtip seems to lack distinct innovation. The author couldn’t compare to existing studies, and the novelty and contribution are not sufficiently articulated. The study feels more like a subtopic suitable for a research project rather than a scientific journal article. Please clearly articulate how this work differentiates itself from existing literature and emphasize its specific innovations.
2. The range of parameters analyzed in this study is narrow, limiting the comprehensiveness of the findings. While numerical simulation methods are particularly effective for exploring a wide range of design parameters and conducting sensitivity analyses, this manuscript does not fully utilize the potential of such tools. Specifically, there is insufficient exploration of the twisting wingtip's actual performance under varied flight conditions. Important parameters, such as wind speed variations, structural stiffness differences, or material properties, are not systematically studied or discussed. A more comprehensive parameter analysis could reveal critical performance trends, provide insights into the robustness of the design, and validate its applicability across broader operational scenarios. Expanding the analysis in these areas would significantly enhance the reliability and impact of the findings.
3. The abstract provides an overview of the study's objectives and methods but lacks specific, quantified outcomes or key findings. It does not clearly present the performance improvements or innovations achieved by the proposed design. Please Include precise quantitative results, such as the percentage reduction in root bending moment, the improvement in lift effectiveness, or the increase in divergence speed, to give readers a clear understanding of the study’s impact and contributions.
Likewise, the conclusion just mentions general advantages of the design, such as improvements in static and dynamic aeroelastic performance, but fails to summarize the study’s key quantitative findings or to provide a clear connection between the results and the research objectives.
4. While the manuscript provides quantitative data, such as the improvement in divergence speed and lift effectiveness, the presentation of these results lacks a systematic summary and in-depth discussion. Key numerical outcomes, including maximum stress values and modal frequencies, are dispersed throughout the text without being cohesively connected to the study’s objectives. Furthermore, there is insufficient comparison between the proposed design and existing technologies, making it challenging to assess the innovation and practical advantages of the proposed twisting wingtip design. Additionally, while figures and tables effectively visualize specific parameters, the manuscript does not adequately discuss the reasoning behind optimal parameter selection (e.g., twisting angle or hinge position) and their implications for real-world applications. The absence of a detailed analysis of how these improvements translate to practical performance limits the impact of the findings.
5. The aeroelastic analysis section presents a framework combining finite element analysis (FEA) and vortex lattice method (VLM) to study static and dynamic aeroelastic behavior. While this approach is theoretically valid and has been used in prior studies, the manuscript does not provide a clear explanation of the methodology, especially regarding how the structural and aerodynamic models are integrated and validated. This lack of clarity is particularly concerning given that traditional aeroelastic analysis often employs more established tools, such as Nastran for comprehensive aeroelastic simulations or iterative coupling of Ansys CFX with structural modeling software for dynamic analysis. Furthermore, while the study introduces key aeroelastic parameters like divergence speed, flutter speed, and lift effectiveness, the analysis remains narrow in scope, with limited parameter exploration and insufficient discussion on how the findings compare to or improve upon existing methods. The authors are encouraged to:
(1). Provide a detailed explanation of the methodology, including model assumptions, coupling techniques, and validation procedures.
(2). Discuss why the chosen approach was preferred over more common alternatives and highlight its advantages or limitations.
Author Response
- The proposed concept of a passive twisting wingtip seems to lack distinct innovation. The author couldn’t compare to existing studies, and the novelty and contribution are not sufficiently articulated. The study feels more like a subtopic suitable for a research project rather than a scientific journal article. Please clearly articulate how this work differentiates itself from existing literature and emphasize its specific innovations.
Answer: Thank you for your suggestions. We have rewritten in many parts and added some results from our studies to make this manuscript have sufficiently novelty.
- The range of parameters analyzed in this study is narrow, limiting the comprehensiveness of the findings. While numerical simulation methods are particularly effective for exploring a wide range of design parameters and conducting sensitivity analyses, this manuscript does not fully utilize the potential of such tools. Specifically, there is insufficient exploration of the twisting wingtip's actual performance under varied flight conditions. Important parameters, such as wind speed variations, structural stiffness differences, or material properties, are not systematically studied or discussed. A more comprehensive parameter analysis could reveal critical performance trends, provide insights into the robustness of the design, and validate its applicability across broader operational scenarios. Expanding the analysis in these areas would significantly enhance the reliability and impact of the findings.
Answer: Thank you for your suggestions. We have rewritten in many parts to make it clearer our research aims, methodology, experimental design, results and discussions, and conclusions.
- The abstract provides an overview of the study's objectives and methods but lacks specific, quantified outcomes or key findings. It does not clearly present the performance improvements or innovations achieved by the proposed design. Please include precise quantitative results, such as the percentage reduction in root bending moment, the improvement in lift effectiveness, or the increase in divergence speed, to give readers a clear understanding of the study’s impact and contributions.
Likewise, the conclusion just mentions general advantages of the design, such as improvements in static and dynamic aeroelastic performance, but fails to summarize the study’s key quantitative findings or to provide a clear connection between the results and the research objectives.
Answer: Thank you for your suggestions. We have addressed all possible suggestions.
- While the manuscript provides quantitative data, such as the improvement in divergence speed and lift effectiveness, the presentation of these results lacks a systematic summary and in-depth discussion. Key numerical outcomes, including maximum stress values and modal frequencies, are dispersed throughout the text without being cohesively connected to the study’s objectives. Furthermore, there is insufficient comparison between the proposed design and existing technologies, making it challenging to assess the innovation and practical advantages of the proposed twisting wingtip design. Additionally, while figures and tables effectively visualize specific parameters, the manuscript does not adequately discuss the reasoning behind optimal parameter selection (e.g., twisting angle or hinge position) and their implications for real-world applications. The absence of a detailed analysis of how these improvements translate to practical performance limits the impact of the findings.
Answer: Thank you for your suggestions. We have rewritten in many parts and addressed all possible suggestions.
- The aeroelastic analysis section presents a framework combining finite element analysis (FEA) and vortex lattice method (VLM) to study static and dynamic aeroelastic behavior. While this approach is theoretically valid and has been used in prior studies, the manuscript does not provide a clear explanation of the methodology, especially regarding how the structural and aerodynamic models are integrated and validated. This lack of clarity is particularly concerning given that traditional aeroelastic analysis often employs more established tools, such as Nastran for comprehensive aeroelastic simulations or iterative coupling of Ansys CFX with structural modeling software for dynamic analysis. Furthermore, while the study introduces key aeroelastic parameters like divergence speed, flutter speed, and lift effectiveness, the analysis remains narrow in scope, with limited parameter exploration and insufficient discussion on how the findings compare to or improve upon existing methods. The authors are encouraged to:
(1). Provide a detailed explanation of the methodology, including model assumptions, coupling techniques, and validation procedures.
(2). Discuss why the chosen approach was preferred over more common alternatives and highlight its advantages or limitations.
Answer: Thank you for your suggestions. We have addressed of your concerns.

Reviewer 2 Report
Comments and Suggestions for Authors
This paper conducts a series of simulation studies on the aeroelastic and structural behavior of twisting wingtips on UAV wings, demonstrating the unique deformation capability and aerodynamic performance of the twisted wing design. However, the following issues exist in this work:
1. This study proposes a conceptual design of a TWT integrating a twisting shaft and employs the TWT twisting wingtip as a passive morphing structure. However, this type of structure appears to be widely used already. This study does not seem to clearly address the significant advantages of the proposed structure compared to previous works utilizing similar designs.
2. Are there any design standards for wing parameters? After determining the overall wing dimensions, parameters of different components such as the twisting shaft and TWT can significantly influence the overall wing's twisting performance. How were these parameters determined in this study? Or, how was it ensured that the selected parameters result in optimal performance?
3. Passive twisting wing deformation is simpler and more adaptive. However, as mentioned in the manuscript, this type of deformation structure is less efficient compared to actively controlled morphing wings. Has this issue been considered in this study? If so, are there any proposed improvements or solutions to address this limitation?
4. The skin is a critical aspect of morphing wing design, as achieving a continuous deformable surface is essential for ensuring the wing's deformation performance. As shown in Figure 1, the morphing wing designed in this study features a two-section structure. When designing the skin structure, was a two-section skin or an integrated skin covering both sections of the wing employed? These two approaches could significantly impact the aerodynamic performance of the wing. If an integrated skin was used, the large twisting angles between the two wing sections could pose challenges to the deformation capabilities of the skin.
5. This paper conducts theoretical modeling and finite element analysis of the wing's structural and aerodynamic performance. Has a prototype been fabricated, and have relevant experiments been designed? Experimental validation using a physical prototype could further verify the accuracy and reliability of the simulation results presented in this study.
6. There are some writing issues in the manuscript. For example, on page 2, line 52, "The concept of PTWThas been illustrated in [7], where a separate rigid wing section, known as the TWT, is mounted at the wingtip through an elastic hinge" should be revised to "The concept of PTWT has been illustrated in [7], where a separate rigid wing section, known as the TWT, is mounted at the wingtip through an elastic hinge". Additionally, the captions for Figures 5 and 6 do not maintain consistent formatting. Please carefully review the manuscript to minimize such errors.
Author Response
This paper conducts a series of simulation studies on the aeroelastic and structural behavior of twisting wingtips on UAV wings, demonstrating the unique deformation capability and aerodynamic performance of the twisted wing design. However, the following issues exist in this work:
- This study proposes a conceptual design of a TWT integrating a twisting shaft and employs the TWT twisting wingtip as a passive morphing structure. However, this type of structure appears to be widely used already. This study does not seem to clearly address the significant advantages of the proposed structure compared to previous works utilizing similar designs.
Answer: Thank you for your suggestions. We have rewritten in many parts to make it clearer different significant and advantages.
- Are there any design standards for wing parameters? After determining the overall wing dimensions, parameters of different components such as the twisting shaft and TWT can significantly influence the overall wing's twisting performance. How were these parameters determined in this study? Or, how was it ensured that the selected parameters result in optimal performance?
Answer: Thank you for your suggestions. We have addressed your suggestions.
- Passive twisting wing deformation is simpler and more adaptive. However, as mentioned in the manuscript, this type of deformation structure is less efficient compared to actively controlled morphing wings. Has this issue been considered in this study? If so, are there any proposed improvements or solutions to address this limitation?
Answer: Thank you for your suggestions. We are sorry for our writing to make it unclear. We have rewritten in the literature review part.
- The skin is a critical aspect of morphing wing design, as achieving a continuous deformable surface is essential for ensuring the wing's deformation performance. As shown in Figure 1, the morphing wing designed in this study features a two-section structure. When designing the skin structure, was a two-section skin or an integrated skin covering both sections of the wing employed? These two approaches could significantly impact the aerodynamic performance of the wing. If an integrated skin was used, the large twisting angles between the two wing sections could pose challenges to the deformation capabilities of the skin.
Answer: Thank you for your concerns. The conceptual design of main body wing and twisting wing tip are separated in two sections with a small gap without covering skin. We have rewritten in aerodynamics modeling part and add more details to make it clearer.
- This paper conducts theoretical modeling and finite element analysis of the wing's structural and aerodynamic performance. Has a prototype been fabricated, and have relevant experiments been designed? Experimental validation using a physical prototype could further verify the accuracy and reliability of the simulation results presented in this study.
Answer: Thank you for your suggestions. Our study is in the conceptual design stage then the reviewer suggestions will study in the 3rd step after we completed consider a reliability-based design to TWT in the next step.
- There are some writing issues in the manuscript. For example, on page 2, line 52, "The concept of PTWThas been illustrated in [7], where a separate rigid wing section, known as the TWT, is mounted at the wingtip through an elastic hinge"should be revised to "The concept of PTWT has been illustrated in [7], where a separate rigid wing section, known as the TWT, is mounted at the wingtip through an elastic hinge". Additionally, the captions for Figures 5 and 6 do not maintain consistent formatting. Please carefully review the manuscript to minimize such errors.
Answer: Thank you for your suggestions. We have proofreading and correct the typos.

Round 2
Reviewer 1 Report
Comments and Suggestions for Authors
The authors have addressed the previously raised concerns and questions by making revisions and clarifications within the manuscript.